# Impact of co-occurring hearing and visual difficulties in childhood on educational outcomes: a longitudinal cohort study

Matilda Hill, [1] Amanda Hall, [2] Cathy Williams, [1] Alan M Emond[1]

► Additional material is published online only. To view, please visit the journal online (http://dx.doi.org/10.1136/bmjpo-2018-000389).

[1]Centre for Child and Adolescent Health, University of Bristol, Bristol, UK
[2]School of Life and Health Sciences, Aston University, Birmingham, UK

**Correspondence to**
Dr Matilda Hill; matildahill84@gmail.com

## ABSTRACT

**Background** Mild hearing and visual difficulties are common in childhood, and both may have implications for educational achievement. However, the impact of co-occurring common hearing and visual difficulties in childhood is not known.

**Objective** To determine the prevalence and impact of co-occurring common hearing and visual difficulties of childhood on educational outcomes in primary and secondary school.

**Methods** The sample was drawn from the Avon Longitudinal Study of Parents and Children, a longitudinal birth cohort study in England. The exposures were hearing and visual difficulties at age 7 (defined as conductive hearing loss or otitis media with effusion, and amblyopia, strabismus or reduced visual acuity, respectively). The outcomes measured were achievement of level 4 or above at Key Stage 2 (KS2) in English, Maths and Science, respectively, at age 11, and attainment of five or more General Certificate of Secondary Education (GCSEs) at grades A*–C at age 16. Multiple logistic regression models assessed the relationship between hearing and visual difficulties and educational outcomes, adjusting for potential confounding factors.

**Results** 2909 children were included in the study; 261 had hearing difficulties, 189 had visual difficulties and 14 children had co-occurring hearing and visual difficulties. Children with co-occurring hearing and visual difficulties were less likely to achieve the national target at KS2 compared with children with normal hearing and vision, even after adjustment for confounding factors (OR 0.30, CI 0.15 to 0.61 for KS2 English). Differences in IQ, behaviour, attention and social cognition did not account for this relationship. The impact of co-occurring hearing and visual difficulties on GCSE results was explained largely by poor performance at KS2.

**Conclusions** Co-occurring hearing and visual difficulties in childhood have an enduring negative impact on educational outcomes. Identification of affected children and early intervention in primary school is essential.

## INTRODUCTION

Hearing and visual difficulties are prevalent among children in the UK. The majority of affected children have mild, temporary difficulties, which may not be formally diagnosed

### What is already known on this topic?

► Mild hearing and visual difficulties are common in childhood.
► The most common cause of hearing difficulties in childhood is chronic otitis media with effusion, which may be associated with lower academic performance.
► The impact of common co-occurring hearing and visual difficulties on educational outcomes is not known.

### What this study hopes to add?

► Children with co-occurring mild hearing and visual difficulties have poorer educational outcomes than children with isolated hearing or visual difficulties.
► Mild visual difficulties alone do not negatively impact on academic performance at school.

and for which they do not receive additional educational support. However, there is evidence that even mild deficits in hearing or vision can have implications for learning and development throughout childhood.[1–5]

The most common cause of chronic hearing loss in childhood is persistent otitis media with effusion (OME, 'glue ear'). There is ongoing controversy regarding the impact of OME on language development and cognition. While several large studies have found no association between OME and language or academic attainment,[6 7] few of these studies have considered concurrent hearing loss or chronicity of OME. Conversely, a number of studies have demonstrated an association between persistent OME-related hearing loss, cognitive development, reading ability and behavioural problems.[2–4 8]

The impact of common visual difficulties on academic attainment in childhood is not well characterised.[9] Decreased visual acuity in childhood is associated with reduced literacy,[1]

but the consequences of more common visual difficulties are not clear. Amblyopia and strabismus are two of the most common visual difficulties encountered in childhood,[10] which often occur in the context of conditions which may in themselves affect development and educational achievement, such as prematurity and low birth weight[11 12]; thus, rigorous adjustment for confounding factors is required to establish the functional impact of these visual difficulties in of themselves.

The impact of co-occurring common hearing and visual difficulties on educational outcomes in childhood has not previously been studied. However, it is well established that integration of auditory and visual information is essential for many cognitive processes, in particular speech perception, and key for the development of language and communication skills.[13 14] A deficit in one of these domains in childhood may therefore hinder developing literacy and render an individual vulnerable to the cumulative effect of an additional sensory deficit.

The aim of this study is to investigate the prevalence and impact of co-occurring common hearing and visual difficulties in childhood on educational outcomes at primary school and secondary school using a large population cohort. We hypothesise that children with co-occurring mild hearing and visual difficulties are more likely to have poorer language and communication skills, and thus lower levels of academic attainment, relative to those with a single sensory deficit or children with normal hearing and vision.

## METHODS
### Study participants
The Avon Longitudinal Study of Parents and Children (ALSPAC) is a longitudinal, population-based birth cohort study. All pregnant women residents in Avon, UK, with expected dates of delivery between 1 April 1991 and 31 December 1992 were eligible for participation, resulting in an enrolment of over 14 000 live births.[15 16]

Please note that the study website contains details of all the data that are available through a fully searchable data dictionary and variable search tool: http://www.bristol.ac.uk/alspac/researchers/our-data/.

### Patient and public involvement
ALSPAC participants advise on ALSPAC studies through the original cohort advisory panel and contribute to the ALSPAC ethics and law committee. ALSPAC participants were not directly involved in the design of this study.

### Hearing and vision assessments
All participating children in ALSPAC were invited to attend a research clinic at 7 years of age. Of the children, 59.3% (8299 children) attended during the period September 1998–September 2000, of whom 98.9% (8205 children) were eligible for inclusion (see figure 1). The clinics included a comprehensive assessment of vision and hearing, the details of which are included in appendix A of the online supplementary material.

### Definition of hearing difficulties
Hearing difficulties were defined as the presence of mild-moderate conductive hearing loss and/or OME in either ear, characterised by air conduction greater than 20 dB and less than 70 dB averaged across 500 Hz, 1 kHz, 2 kHz and 4 kHz (based on the British Society of Audiology definitions[17]), or the presence of a type B tympanogram, respectively. The prevalence of OME decreases with age after the first 2 years of life[18]; children with evidence of OME at age 7 are likely to have persistent OME, which is associated with conductive hearing loss in over 70% of cases.[19 20] However, hearing loss may be fluctuant and thus not captured by a single clinic assessment, hence the utilisation of both air conduction tests and tympanometry to identify children with hearing difficulties.

A total of five children with sensorineural hearing loss, defined by bone conduction greater than 30 dB at either 1 kHz or 4 kHz, were excluded from this analysis.

### Definition of visual difficulties
Children with 'clinically significant' strabismus, amblyopia or mild-moderate reduced acuity (based on the WHO International Classification of Diseases-11 definition[21]) were defined as having visual difficulties. 'Clinically significant' strabismus comprised all children with manifest strabismus or previously defined large latent deviations (≥10 prism dioptre if convergent and ≥15 prism dioptre if divergent).[10] Amblyopia was defined as a history of patching treatment and/or an interocular difference in acuity of >0.2 logarithm of the minimum angle of resolution (logMAR) units, where the worst-seeing eye had an acuity of >0.3 logMAR. Reduced acuity was defined by reduced distance acuity of the better-seeing eye ≥0.3 logMAR. Acuity was assessed with glasses if worn ('habitual' state), and in the 'habitual state plus pinhole', as a proxy for full refractive correction.

Refractive errors were not included in our definition of visual difficulties as these are potentially correctable with glasses and have already been studied extensively in the context of educational achievement, with myopia being linked with higher educational achievement.[22 23]

Nineteen children with known ocular pathology or severe visual impairments (>1.0 logMAR) were excluded. Triplets, quadruplets and children with Down's syndrome or cerebral palsy were also excluded from this analysis.

### Educational outcomes
Educational outcomes at primary school were assessed using Standardised Assessment Test results, obtained from the National Pupil Database (NPD).[24] Key Stage 2 (KS2) tests are undertaken during the final year of primary school (year 6) at age 10–11. The national expected standard is achievement of national curriculum level 4 or above, and we therefore used achievement of level ≥4 at KS2 in English, Maths and Science as our three

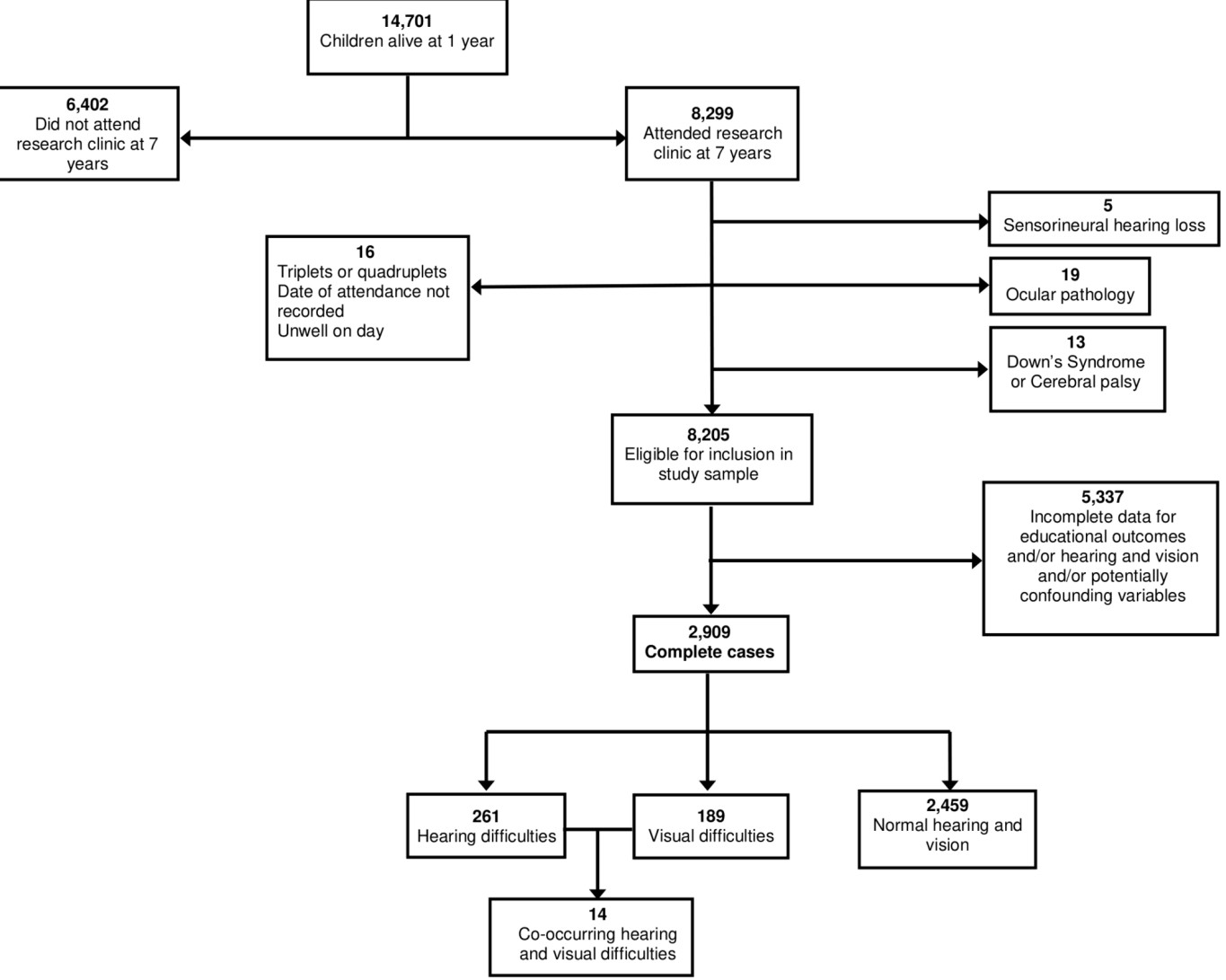

**Figure 1** Study sample flow chart showing the inclusion and exclusion criteria.

educational outcomes in primary school; performance in each subject was analysed separately.

Educational outcomes in secondary school were assessed using General Certificate of Secondary Education (GCSE) results at Key Stage 4 (KS4). GCSEs are taken at the end of compulsory schooling (year 11) at age 16 and are graded A*–G. Achievement of 5 or more GCSEs (including Maths and English) at grades A*–C is the national benchmark measure of achievement, and we used this as our single outcome at secondary school. We adjusted outcomes for KS2 attainment, as performance at KS2 is well known to predict performance at GCSEs.[25]

Information on children receiving Special Educational Needs (SEN) support was provided by the Pupil Level Annual School Census (PLASC). SEN status was obtained for each child at KS2 and KS4, with information on the level of individual educational support being provided. We dichotomised these data into children receiving no special provision and those receiving some level of support.

NPD and PLASC data were not available for children attending independent schools, or schools outside of England, and these children were excluded from this analysis.

### Confounding and mediating factors
Gender, prematurity, low birth weight, admission to a special care baby unit (SCBU), maternal age, parity, smoking during pregnancy, duration of breast feeding and socioeconomic status were selected as potential confounding factors, based on their previously identified associations with hearing or visual difficulties in childhood and well-established links with academic performance.[10–12 26–29]

The ALSPAC Family Adversity Index, derived from a questionnaire about socioeconomic status completed by mothers at 2–4 years, was used as a measure of family adversity. The Indices of Multiple Deprivations at age 7, a census-derived score of relative deprivation of a neighbourhood, was used as a measure of social deprivation.

IQ, attention, social cognition and behaviour were selected as potential mediating factors, as they may be influenced by visual and hearing difficulties and can impact on educational achievement.[1 3 4 8] Reading ability was not included due to high rates of missing data. These domains were tested during a research clinic to which all participating children were invited and 7488 children (53.5%) attended at 8 years of age. Further information on these assessments is provided in appendix B of the online supplementary material.

### Analyses

Binary univariate logistic regressions were used to calculate ORs and 95% CI to assess the relationship between hearing and visual difficulties and educational outcomes. The analyses were repeated, controlling for all potential confounding factors significant at the 5% level in the univariate analyses and all potential mediating factors which fulfilled the Baron and Kenny mediation model steps 1–3.[30] These multiple logistic regression models sequentially adjusted for individual factors, maternal factors, wider socioeconomic factors, earlier educational performance, IQ and additional mediating factors. Further information is provided in tables 2 and 3.

Multiple imputation using chained equations was used to impute missing data for all variables included in the final logistic regression models, including the outcome, and variables that predicted missingness. This technique helps to minimise attrition bias and improve precision of estimates.[31] We imputed data for all 8205 children who attended the research clinic at 7 years and did not meet the exclusion criteria. Twenty imputations were performed. All analyses were carried out using STATA V.15.0. Further information regarding the multiple imputation is provided in appendix C of the online supplementary material.

### RESULTS
#### Sample characteristics
Of the 8205 children who attended the research clinic at 7 years and were eligible for inclusion, 2909 (33.5%) had complete data for hearing and vision, educational outcomes and potential confounding variables (see figure 1).

Children attending the research clinic were more likely to come from families of higher socioeconomic class and achieve higher levels of academic attainment at both KS2 and KS4 than children who did not attend the clinic (see online supplementary material, table ST2).

#### Prevalence and causes of hearing and visual difficulties
Of the 2909 children with complete data, 261 (9.0%) had hearing difficulties, 189 (6.5%) had visual difficulties and 14 (0.5%) had both hearing and visual difficulties. Types of hearing and visual difficulties are demonstrated in table 1.

Conductive hearing loss was identified in 159 children; 145 of these children (91%) had mild hearing loss (air conduction of 21.25–40 dB, mean 27.68 dB), and 14 children had moderate hearing loss (air conduction of 41.25 dB–66.25 dB, mean 46.25 dB). The majority of children with conductive hearing loss had evidence of concurrent OME.

The overall prevalence of OME in either or both ears was 6.7%. A higher proportion of children undergoing assessment in winter months had OME compared with those attending clinic in the summer months (8.0% vs 5.6%), but this association was weak (p=0.09).

Amblyopia was the most common cause of visual difficulties, affecting 4.4% of all children. There was considerable overlap between amblyopia and strabismus; of the 128 children with amblyopia, 47 (36.7%) also had strabismus.

Only seven children had reduced habitual acuity in the best-seeing eye; four children had mildly reduced visual acuity (0.3–0.44 logMAR), and three children had moderately reduced acuity (0.50–0.7 logMAR), with a mean acuity of 0.45 logMAR. These reduced acuities most likely represent uncorrected refractive errors, given that acuity was not assessed with full refractive correction and the exclusion of children with ocular pathology.

| Table 1 Types of hearing and visual difficulties | |
| --- | --- |
| **Types of hearing and visual difficulties** | **Proportion of children with either hearing or visual difficulties (%)** |
| Hearing difficulties | 261 (100) |
| Otitis media with effusion | 102 (39.1) |
| Otitis media with effusion and conductive hearing loss | 94 (36.0) |
| Conductive hearing loss | 65 (24.9) |
| Visual difficulties | 189 (100) |
| Amblyopia | 79 (41.8) |
| Strabismus | 56 (29.6) |
| Strabismus and amblyopia | 47 (24.9) |
| Reduced acuity with or without amblyopia and/or strabismus | 7 (3.7) |

**Table 2** Multiple regression models using the imputed data set (n=8205) demonstrating the likelihood children achieved level ≥4 at KS2 English according to hearing and visual status and after adjustment for confounding variables (models 1–3) and mediating factors (models 4–5)

KS2 English: achievement of level ≥4

| | Univariate logistic regression | | Model 1: male, age at testing and SEN at KS2 | | Model 2: model 1 and maternal age <25, parity and smoking during pregnancy | | Model 3: model 2 and IMD | | Model 4: model 3 and FAI | | Model 5: model 4 and behaviour, social perceptions and attention | |
|---|---|---|---|---|---|---|---|---|---|---|---|---|
| | OR | 95% CI | OR | 95% CI | OR | 95% CI | OR | 95% CI | OR | 95% CI | OR | 95% CI |
| Hearing difficulties (n=852) | 0.78 | 0.66 to 0.92 | 0.80 | 0.66 to 0.98 | 0.81 | 0.66 to 0.99 | 0.81 | 0.66 to 0.99 | 0.87 | 0.71 to 1.06 | 0.88 | 0.72 to 1.08 |
| Visual difficulties (n=573) | 0.66 | 0.64 to 0.80 | 0.79 | 0.62 to 0.99 | 0.81 | 0.64 to 1.02 | 0.82 | 0.64 to 1.03 | 0.87 | 0.68 to 1.11 | 0.90 | 0.71 to 1.14 |
| Co-occurring hearing and visual difficulties (n=45) | 0.25 | 0.14 to 0.45 | 0.30 | 0.15 to 0.61 | 0.30 | 0.15 to 0.61 | 0.30 | 0.15 to 0.61 | 0.36 | 0.17 to 0.74 | 0.38 | 0.18 to 0.78 |

FAI, Family Adversity Index; IMD, Indices of Multiple Deprivations; KS2, Key Stage 2; SEN, Special Educational Needs.

**Table 3** Multiple regression models using the imputed data set (n=8205) demonstrating the likelihood children achieved ≥5 GCSEs at A*–C grade (including Maths and English), according to hearing and visual status and after adjustment for confounding variables (models 1–3) and mediating factors (models 4–5)

KS4: achievement of ≥5 GCSEs at A*–C, including Maths and English

| | Univariate logistic regression | | Model 1: male, SCBU admission and SEN at KS4 | | Model 2: model 1 and maternal age <25, parity and smoking during pregnancy | | Model 3: model 2 and FAI and IMD | | Model 4: model 3 and KS2 English, Maths and Science performance | | Model 5: model 4 and IQ | | Model 6: model 5 and behaviour, social perceptions and attention | |
|---|---|---|---|---|---|---|---|---|---|---|---|---|---|---|
| | OR | 95% CI | OR | 95% CI | OR | 95% CI | OR | 95% CI | OR | 95% CI | OR | 95% CI | OR | 95% CI |
| Hearing difficulties (n=852) | 0.80 | 0.68 to 0.94 | 0.83 | 0.69 to 0.99 | 0.83 | 0.69 to 0.99 | 0.82 | 0.68 to 0.99 | 0.85 | 0.69 to 1.04 | 0.95 | 0.76 to 1.17 | 0.96 | 0.77 to 1.19 |
| Visual difficulties (n=573) | 0.86 | 0.71 to 1.05 | 0.94 | 0.74 to 1.17 | 1.00 | 0.79 to 1.27 | 1.02 | 0.81 to 1.30 | 1.20 | 0.92 to 1.56 | 1.34 | 1.00 to 1.79 | 1.37 | 1.02 to 1.84 |
| Co-occurring hearing and visual difficulties (n=45) | 0.55 | 0.29 to 1.06 | 0.67 | 0.29 to 1.51 | 0.68 | 0.29 to 1.58 | 0.66 | 0.28 to 1.53 | 1.13 | 0.43 to 2.94 | 1.42 | 0.50 to 4.06 | 1.49 | 0.52 to 4.29 |

FAI, Family Adversity Index; GCSE, General Certificate of Secondary Education; IMD, Indices of Multiple Deprivations; KS2, Key Stage 2; KS4, Key Stage 4; SCBU, special care baby unit; SEN, special educational needs.

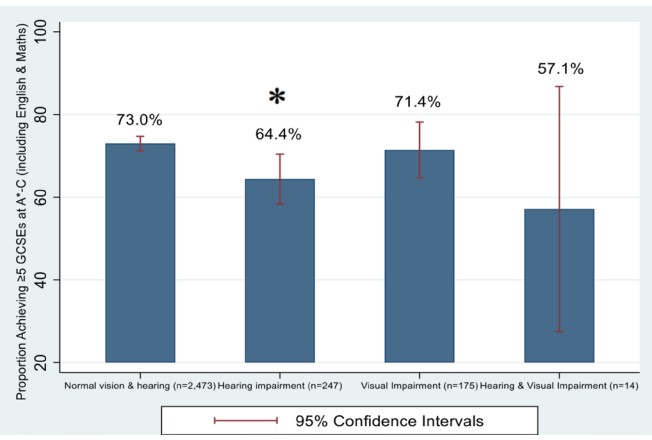

**Figure 2** Graph showing the proportion of students achieving ≥5 GCSEs at A*–C (including Maths and English) by hearing and vision status. GCSE, General Certificate of Secondary Education. * denotes significance at p<0.05 level.

Children with hearing and visual difficulties had similar characteristics to those with normal hearing and vision (see online supplementary material, table ST3). However, children with visual difficulties were more likely to have mothers who smoked during pregnancy (24.6% vs 18.0%, p=0.03), and children with hearing difficulties were more likely to have been admitted to SCBU in infancy (9.3% vs 5.4%, p=0.01).

### Educational outcomes and SEN status in primary school
The proportion of children in this study achieving level 4 or above at KS2 exceeded the national average; 92.5% and 97.7% of children achieved level 4 or above in KS2 English and Maths tests, respectively, compared with 75% and 73% of children nationally.[32] The proportions of children achieving these targets by hearing and visual status are shown in online supplementary material, table ST4.

Children with either hearing or visual difficulties were less likely to achieve level 4 or above at KS2 English tests relative to children with normal hearing and vision (see table 2 for the imputed data set analysis; see online supplementary material, table ST8–10 for the complete cases analysis). However, these relationships were attenuated by adjustment for confounding factors, and further attenuated after adjustment for IQ, suggesting that some of the association is mediated through IQ. In contrast, children with co-occurring hearing and visual difficulties remained less likely to achieve this target after controlling for confounding factors (OR 0.30, CI 0.15 to 0.61). This relationship was not significantly altered by adjustment for IQ or other potential mediating factors.

The relationship is similar for performance in KS2 Maths and KS2 Science tests, although hearing difficulties appear to have less of an impact (see online supplementary material, table ST6 and ST7).

The proportion of children with visual difficulties receiving SEN support was higher compared with children with normal hearing and vision (11.4% vs 6.2%, p=0.02; see online supplementary material, table ST5).

Overall, children with SEN status at KS2 were less likely to achieve level 4 or above in KS2 English, Maths and Science tests than those without formal support (52.6% vs 87.6%, p=0.00).

### Educational outcomes and SEN status in secondary school
Overall, almost three-quarters (72.1%) of the study sample achieved five or more GCSEs (including English and Maths) at A*–C grade, which is significantly higher than the national average of 47.6% children achieving this standard the same year.[33] The proportion of children with hearing difficulties achieving this target was lower compared with those with normal hearing and vision (see figure 2).

The association between hearing difficulties and poorer performance at GCSEs is attenuated after adjustment for performance at KS2 and IQ, suggesting that the association between hearing and education is mediated through these factors (see table 3 for the imputed data set analysis; see online supplementary material, table ST11 for the complete cases analysis). Children with visual difficulties alone were no less likely to attain five or more GCSEs at A*–C; they were in fact more likely to achieve this target after adjustment for confounding variables and IQ.

Children with co-occurring hearing and visual difficulties were less likely to achieve the national target even after adjustment for confounding factors, although this association was weak (OR 0.66, CI 0.28 to 1.53). This relationship is attenuated after adjustment for performance at KS2, suggesting that poorer outcomes at KS4 in these children can be partly explained by poorer educational outcomes at primary school.

At KS4, the overall proportion of children receiving SEN support was similar to that at KS2 (6.7% vs 6.5%, p=0.76). Children with visual difficulties were almost twice as likely to receive SEN support than those with normal hearing and vision (see online supplementary material, table ST5). Children with SEN provision were less likely to attain five GCSEs at A*–C (32.3% vs 74.9%, p=0.00).

### DISCUSSION
#### Discussion of results
In this longitudinal study we have demonstrated that co-occurring mild hearing and visual difficulties in childhood have a negative impact on educational outcomes, greater than the effect of hearing or visual difficulties alone. We have shown a weak association between mild hearing difficulties at age 7 and academic achievement at KS2 and KS4, which may be mediated by IQ. We found no negative association between mild visual difficulties and academic outcomes. In contrast, children with co-occurring hearing and visual difficulties were less likely to attain the expected academic standards at the end of primary school, an effect which is not wholly mediated through IQ, behaviour, attention or social cognition, suggesting a substantial, unexplained educational

disadvantage. They were also less likely to achieve five or more GCSEs at KS4, which may be attributable to poor performance at KS2. However, this relationship is weak, likely due to the relatively low prevalence of co-occurring mild sensory difficulties and potential small effect size.

A higher proportion of children with visual difficulties had SEN status at both primary school and secondary school relative to those with normal hearing and vision. Common visual difficulties, as defined by this study, are of themselves not sufficient to merit SEN support, suggesting a higher prevalence of additional disabilities among children with visual difficulties. This is consistent with previously reported findings.[34] Children receiving SEN support are less likely to achieve the national targets at both primary school and secondary school compared with those without SEN support, implying SEN status is a proxy indicator of severity of educational difficulties.

### Strengths and weaknesses

The major strength of this study is that it uses a large population-based birth cohort, with an accurate assessment of hearing and vision. Furthermore, it uses standardised national tests as objective outcome measures, and longitudinally collected data on a wide range of confounders.

The most important limitation of this study relates to missing data and the under-representation of children from ethnic minorities and those from lower socioeconomic backgrounds. A multiple imputation technique was used to minimise potential bias and improve precision. An additional limitation is the single time-point used to assess hearing, and possible inclusion of children with transient OME as a result, which may have led to underestimation of the effect of persistent OME-related hearing loss on academic attainment. Furthermore, we did not have information regarding which children were treated for their hearing and visual difficulties either prior or following the assessment at age 7. Finally, while we excluded children with known diagnoses of Down's syndrome or cerebral palsy, we did not have full data regarding developmental disorders or medical comorbidities. However, such conditions affect only a small minority of children with hearing or visual difficulties, and this was a population-based study designed to evaluate the impact of common, mild difficulties.[12 35]

### In the context of other research

Previous studies investigating the impact of co-occurring hearing and visual difficulties in childhood have focused on children with severe deficits. Developmental disorders commonly co-occur with severe hearing and visual difficulties, or 'deaf-blindness',[34–36] which often precludes children from attending mainstream schools and has additional implications for their education. To our knowledge, our study is the first to investigate the impact of co-occurring common mild hearing and visual difficulties on academic attainment, in a population of children attending mainstream schools.

The impact of childhood OME on cognition and educational performance has been studied previously by prospective cohort studies, although there is no clear consensus. Data from the longitudinal Dunedin Study demonstrated an association between childhood OME and lower IQ in early teenage years,[2] which is consistent with findings from the Aarhus Birth Cohort which reported hearing loss at 9–11 years was associated with behavioural problems and reading difficulties.[8] In contrast, data from the Danish National Birth Cohort showed no association between childhood OME episodes and school performance, findings supported by a meta-analysis of prospective studies which found little to no association between OME and speech and language development.[6 7] The disparity in these findings is likely explained by the heterogeneity in study design, including inconsistencies in adjustment for confounding variables and consideration of associated hearing loss and persistence of OME. Our findings demonstrated a small effect of OME and/or hearing loss at age 7 on academic performance at age 10–11 and 16 years after adjustment for confounding variables, mediated through IQ. This is consistent with data from the 1970 British Birth Cohort[3] and previously published work using ALSPAC data, which have shown an association between OME and lower IQ,[4] although their results suggest this association diminishes with age.

Previous research investigating the impact of common visual difficulties in childhood on educational outcomes has focused largely on refractive errors, which were not included in this study.[22 23] There is, however, convincing evidence from a cross-sectional study involving participants of the Born in Bradford birth cohort study that reduced visual acuity at age 4–5 years is associated with reduced literacy.[1] However, the prevalence of reduced visual acuity was considerably lower in our study (<1% vs 4%, likely due to the different socioeconomic and ethnic demographics of the cohorts). Data from the 1958 British Birth Cohort demonstrated no impact of unilateral amblyopia in childhood on educational tests at age 7, 11 and 16, or on highest educational qualification obtained.[37] Amblyopia was the most common cause of visual difficulties in our study, and our findings are consistent with this.

Furthermore, we demonstrated higher rates of SEN among children with visual difficulties; this is consistent with previous research demonstrating that visual difficulties commonly co-occur with developmental disorders.[35] We are, however, unable to explain the observed positive effect of visual difficulties on performance at secondary school after adjustment for confounders and IQ, a finding that has been reported in the context of refractive errors previously.[22]

### Clinical and research implications

This study has important clinical implications. Children with known visual or hearing difficulties should be routinely tested for additional sensory difficulties, as even

mild co-occurring visual or hearing deficits are associated with poorer educational outcomes. The impact of co-occurring hearing and visual difficulties on performance at secondary school is largely explained by poor performance at primary school; hence, early identification and intervention is essential.

Future research involving larger numbers of participants are required to replicate these findings and elucidate further the factors mediating this association. We recommend that future investigators focus next on the role of reading and language skills as potential mediators which could explain these findings.

**Acknowledgements** We are extremely grateful to all the families who took part in this study, the midwives for their help in recruiting them, and the whole ALSPAC team, which includes interviewers, computer and laboratory technicians, clerical workers, research scientists, volunteers, managers, receptionists and nurses. The authors thank Dr Linda Hollen for providing advice on the statistical methodology.

**Contributors** The study was designed jointly by MH, AME, AH and CW. MH carried out the data analysis and wrote the initial draft. AME, AH and CW contributed equally to revisions of the draft.

**Funding** The UK Medical Research Council, the Wellcome Trust (grant ref: 102215/2/13/2) and the University of Bristol provide core support for ALSPAC. A comprehensive list of grants funding is available on the ALSPAC website: http://www.bristol.ac.uk/alspac/external/documents/grant-acknowledgements.pdf.

**Competing interests** None declared.

**Patient consent for publication** Not required.

**Ethics approval** Ethical approval for the study was obtained from the ALSPAC Ethics and Law Committee and the local research ethics committee.

**Provenance and peer review** Not commissioned; externally peer reviewed.

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
