## [Reviewer comments · BMJ Paediatrics Open]

ARTICLE DETAILS

TITLE (PROVISIONAL)	THE IMPACT OF CO-OCCURRING HEARING AND VISUAL DIFFICULTIES IN CHILDHOOD ON EDUCATIONAL OUTCOMES: A LONGITUDINAL COHORT STUDY
AUTHORS	Hill, Matilda; Hall, Amanda; Williams, Cathy; Emond, Alan

VERSION 1 – REVIEW

REVIEWER	Reviewer name: Peter Flom Institution and Country: Peter Flom Consulting, USA Competing interests: None
REVIEW RETURNED	20-Oct-2018

GENERAL COMMENTS	I confine my remarks to statistical aspects of this paper. In the abstract, it's unclear how the outcomes are dealt with. Is it 4 different outcomes? Or a count of the successes? In either case, the analysis can't be "multivariate regression" it is either multivariate logistic regression (with 4 dichotomous outcomes) or multiple count regression (if it's a count of successes). This is also not clear on page 8: "We used achievement of level 4 at KS2 English, Maths and Science as our educational outcomes in primary school". Did they have to achieve at level 4 on all three? Or any of the three? Or was it 3 separate analyses? For the secondary school students, it looks like a single variable was used. This all becomes a little clearer in the results section, but it should be explained here. For both sets of students, the result was dichotomized. This is almost always a bad idea. It lessens statistical power and introduces a sort of "magical thinking" that something suddenly changes at a cutoff point (e.g. 5 GCSEs). It's not exactly *wrong* to do this, but it's not good. It would be better to use, perhaps, count of GCSEs in secondary school. "Multivariate XXX regression" usually means that there are multiple dependent variables; when there are multiple independent variables, the better term is "Multiple XXX regression". Page 9 - the whole analysis section is quite unclear. Does the first line of this section refer to differences in variables other than the main ones? If so, which variables? If not, then they shouldn't be used because they don't account for other variables. Mediation analysis is usually done in a more comprehensive way than seems to be the case here. It's true that one aspect of mediation is a reduction in the size of parameter estimates. But this doesn't distinguish mediation from confounding.
---

	Two typical approaches to mediation are 1) that of Baron and Kenny 2) That of MacKinnon. Missing data - for MICE (or any imputation method) it is important to assess the reason for missingness, especially when so much data is missing. Was it missing completely at random? Missing at random? Missing not at random? Why was data imputed for all children when some of them had complete data? Page 15 - the results for secondary students weren't just attenuated, they were reversed! Students with visual impairments or both sorts of impairments actually did BETTER than those without (ORs are over 1) and this was significant for those with only visual impairments. This surely needs discussion. I think there's a lot of interesting stuff here, but it needs considerable revision before I can recommend publication.
--	--

REVIEWER	Reviewer name: Katherine M Spowart Institution and Country: Greater Glasgow and Clyde NHS Board, UK Competing interests: None
REVIEW RETURNED	02-Nov-2018

GENERAL COMMENTS	There is I am afraid a fundamental problem with the definition of the study population in this paper and in the terminology used. Visual /hearing Impairment and difficulty are used interchangeably throughout but there is a clear difference. The author is reporting on the impact of hearing and visual difficulties NOT impairments. A mild hearing loss is defined internationally as a loss of $> \text{ or } = 25\text{dB}$ (there is no reason given why the author included 20dB, generally regarded as normal hearing) and in no definition is amblyopia or squint regarded as a visual impairment (96% of those identified fall into this group). The cut-off in terms of visual acuity is debated with a shift from ICD 10 to ICD11 but it is clear that the level is poorer than 0.300 whereas those with an acuity of 0.300 are included in this study. This devalues the results as the study population is unclear. More information on the levels of hearing loss and reduced visual acuity found would be helpful to clarify the target population . It is unclear what the range of visual acuity and hearing loss found was and this is important as it is already known, as stated, that hearing impairment and visual impairment lead to poorer educational outcomes. 5 children were excluded due to SN hearing loss but no information is given on thresholds for exclusion on the grounds on visual acuity. The study should therefore concentrate on those with acuities of 0.325 - 0.475 (not generally accepted as visual impairment), and those with hearing loss in the range 25-40dB (not generally treated as hearing impairment.) Although this may reduce the small numbers reported already, it should address the research question better and help to indicate if there is truly evidence that mild visual and hearing difficulty may increase negative educational outcome. It would also be important to reference the reason for the thresholds chosen for attainment at KS2 and KS4 and include the normal distribution for this to compare with the study population , particularly given the differences in population from other studies as outlined in the text.
---

	Whilst the author has included in figure1 the information demonstrating numbers included / excluded from original cohort it would be helpful for clarity to extend this to demonstrate the numbers found to fulfil the visual/hearing criteria . There is a wealth of statistical data included particularly within the supplementary material which perhaps could be reduced, concentrating on the key information on the target group rather than justification of the use of regression models to increase numbers which personally I find unhelpful. I do think this group an important one to study as they could represent a potentially neglected group within education. I would suggest that the definition of the study population be revisited , be more clearly defined with justification included if it varies from accepted norms, and that the term impairment is not used. The analysis should concentrate on the actual group studied rather than using imputation which, given the unknown variables and timescales involved I would suggest may not be entirely reliable. In view of the above, I find the conclusion that the results suggest a 'substantial' educational disadvantage at present not fully evidenced.
--	--

REVIEWER	Reviewer name: Catherine Tuffrey Institution and Country: Solent NHS Trust, UK Competing interests: None
REVIEW RETURNED	15-Nov-2018

GENERAL COMMENTS	This is an important topic which affects a significant number of children in the UK. The findings should be of concern to families and educational services as well as being important for clinical teams to be aware of considering assessment as well as when communicating with families and other professionals. A clearly written paper. I am not an expert in statistics and cannot comment on the appropriateness or otherwise of statistics used here. There are a few areas where I feel additional explanation might be beneficial for the non-expert reader: Missing data - although it is in the supplementary material, the main paper does not appear to mention the response rate for the clinics at 7 years (or indeed for those at 8, and 9 years) where the data reported here was collected. This would be helpful to have in the main paper. Linked to this is degree of missing data. It is acknowledged that longitudinal studies of this type will inevitably have high attrition over the long term, and missing data from later visits such as the cognitive tests and SDQ data I understand. However, it would be helpful if the authors could explain why data such as ethnicity which presumably could be reasonably easily collected at each visit was missing even when the family attended the 7 year clinic. (Maybe parent choice played a part, but could you tell us?) In the introduction, it is stated that 'We hypothesise that children with co-occurring mild hearing and visual impairments are more likely to have poorer language and communication skills, and thus lower levels of academic attainment, relative to those with a single sensory deficit or children with normal hearing and vision.' However, the paper does not mention specific measures of language and communication skills as far as I can see, and this is not referred to again.
---

	I wonder whether the measurement of such possible mediating factors could be mentioned in the discussion of future research directions? Might there be other possible causes such as teacher expectations (which the educational literature suggests can affect outcomes) or absences from school, which might lead to these findings? If you need to shorten the paper, I wonder how much the general reader will gain from the discussion of imputation of missing values, and some of this could be included in the supplementary material?
--	--

VERSION 1 – AUTHOR RESPONSE

Reviewer 1:

Comment: In the abstract, it's unclear how the outcomes are dealt with. Is it 4 different outcomes? Or a count of the successes? In either case, the analysis can't be "multivariate regression" it is either multivariate logistic regression (with 4 dichotomous outcomes) or multiple count regression (if it's a count of successes).

Response: We used 4 different outcomes. We have amended the Abstract to clarify this.
 "The outcomes measured were achievement of level 4 or above at Key Stage 2 (KS2) in English, Maths and Science, respectively, at age 11, and attainment of 5 or more GCSEs at grades A*-C at age 16."

We have changed the phrase multivariate regression to multivariate logistic regression throughout.

Comment: This is also not clear on page 8: "We used achievement of level 4 at KS2 English, Maths and Science as our educational outcomes in primary school". Did they have to achieve at level 4 on all three? Or any of the three? Or was it 3 separate analyses?
 For the secondary school students, it looks like a single variable was used. This all becomes a little clearer in the results section, but it should be explained here.

Response: We have amended the Methods to clarify that we performed 3 separate analyses for primary school outcomes, and a single analysis for secondary school outcomes.
 "The national expected standard is achievement of National Curriculum level 4 or above, and we therefore used achievement of \geq level 4 at KS2 in English, Maths and Science as our three educational outcomes in primary school; performance in each subject was analysed separately."

"Achievement of 5 or more GCSEs (including Maths and English) at grades A*- C is the national benchmark measure of achievement, and we used this as our single outcome at secondary school."

Comment: For both sets of students, the result was dichotomized. This is almost always a bad idea. It lessens statistical power and introduces a sort of "magical thinking" that something suddenly changes at a cutoff point (e.g. 5 GCSEs). It's not exactly *wrong* to do this, but it's not good. It would be better to use, perhaps, count of GCSEs in secondary school.

Response: We chose to dichotomise the data for both primary school and secondary school results because achievement of $>$ level 4 in KS2 & achievement of >5 GCSEs at A*-C (incl. Maths & English) are both used nationally as benchmark measures of academic achievement. Educational outcomes are commonly assessed and compared based on achievement of these nationally-set standards.

We have amended the Methods to explain this more clearly.

“The national expected standard is achievement of National Curriculum level 4 or above, and we therefore used achievement of \geq level 4 at KS2 in English, Maths and Science as our three educational outcomes in primary school; performance in each subject was analysed separately”

“Achievement of 5 or more GCSEs (including Maths and English) at grades A*- C is the national benchmark measure of achievement, and we used this as our single outcome at secondary school.”

Comment: "Multivariate XXX regression" usually means that there are multiple dependent variables; when there are multiple independent variables, the better term is "Multiple XXX regression".

Response: We have changed the phrase multivariate logistic regression to multiple logistic regression throughout the paper.

Comment: Page 9 - the whole analysis section is quite unclear. Does the first line of this section refer to differences in variables other than the main ones? If so, which variables? If not, then they shouldn't be used because they don't account for other variables.

Response: We accept that the opening line of the Analyses section was unclear and have amended this section. The student's t-test and Pearson's chi squared test were used to compare demographic characteristics of children that attended clinic vs those that did not, and the proportions of children receiving SEN support by hearing and vision status, respectively. This data is presented in the Supplementary Material, which we have amended to include details of the statistical tests used in the relevant table descriptions.

Comment: Mediation analysis is usually done in a more comprehensive way than seems to be the case here. It's true that one aspect of mediation is a reduction in the size of parameter estimates. But this doesn't distinguish mediation from confounding. Two typical approaches to mediation are 1) that of Baron and Kenny 2) That of MacKinnon

Response: We acknowledge that we did not provide enough information regarding the mediation model used.

We used the Baron & Kenny method to demonstrate:

- 1) Hearing & visual status is related to educational outcomes
- 2) Hearing & visual status is related to IQ, attention, social cognition & behaviour
- 3) IQ, attention, social cognition & behaviour is related to educational outcomes after adjusting for hearing & vision
- 4) We were unable to demonstrate complete mediation via these factors however, as shown in the multiple logistic regression models presented in the paper.

We have not presented all 4 steps in the paper, as we do not feel many readers would find this helpful. We have, however, amended the Methods section to clarify this.

“The analyses were repeated, controlling for all potential confounding factors significant at the 5% level in the univariate analyses and all potential mediating factors which fulfilled the Baron and Kenny mediation model steps 1-3(29).”

Comment: Missing data - for MICE (or any imputation method) it is important to assess the reason for missingness, especially when so much data is missing. Was it missing completely at random? Missing at random? Missing not at random?

Response: We have amended Appendix C of the Supplementary Material to make this clearer.

“In addition, we identified factors which predicted missingness using logistic regression analyses and imputed these variables (maternal smoking during pregnancy, male gender and maternal age <25 years). All of these variables were included in the final model, thus the assumption of “missing at random” is supported.”

Comment: Why was data imputed for all children when some of them had complete data?

Response: Whilst 2,909 children had complete data for hearing, vision, educational outcomes and potential confounding variables, not all of these children had complete data for potential mediating factors. We therefore chose to impute data for all the children that attended clinic.

Comment: Page 15 - the results for secondary students weren't just attenuated, they were reversed! Students with visual impairments or both sorts of impairments actually did BETTER than those without (ORs are over 1) and this was significant for those with only visual impairments. This surely needs discussion.

Response: We agree that this is an interesting finding, and have elaborated in the Results and Discussion.

“Children with visual difficulties alone were no less likely to attain 5 or more GCSEs at A*-C; they were in fact more likely to achieve this target after adjustment for confounding variables and IQ.”

“We are, however, unable to explain the observed positive effect of visual difficulties on performance at secondary school after adjustment for confounders and IQ; a finding that has been reported in the context of refractive errors previously(21).”

Reviewer 2:

Comment: Visual /hearing Impairment and difficulty are used interchangeably throughout but there is a clear difference. The author is reporting on the impact of hearing and visual difficulties NOT impairments.

Response: We accept that visual and/or hearing “difficulty” is the more appropriate terminology, and have removed the phrase “impairment” from the paper.

Comment: More information on the levels of hearing loss and reduced visual acuity found would be helpful to clarify the target population . It is unclear what the range of visual acuity and hearing loss found was and this is important as it is already known, as stated, that hearing impairment and visual impairment lead to poorer educational outcomes. 5 children were excluded due to SN hearing loss but no information is given on thresholds for exclusion on the grounds on visual acuity. The study should therefore concentrate on those with acuities of 0.325 - 0.475 (not generally accepted as visual impairment), and those with hearing loss in the range 25-40dB (not generally treated as hearing impairment.)

Response: With regards to hearing loss, we used the British Society of Audiologists definition of mild-moderate conductive hearing loss (20-40 dB and 41-70 dB, respectively), which underlie UK clinical guidelines. This is in fact stricter than the US NHANES definition of hearing loss as >15 dB. We have amended the Methods to add clarify this.

“Hearing difficulties were defined as the presence of mild-moderate conductive hearing loss and/or OME in either ear, characterised by air conduction greater than 20 dB and less than 70 dB averaged across 500 Hz, 1kHz, 2kHz and 4kHz (based on The British Society of Audiology definitions - reference), or the presence of a type B tympanogram, respectively.”

We agree that providing further information on the levels of hearing loss identified is helpful to the reader and have included this information in the Results section.

“Conductive hearing loss was identified in 159 children; 145 of these children (91%) had mild hearing loss (air conduction of 21.25 – 40 dB, mean 27.68dB), and 14 children had moderate hearing loss (air conduction 41.25dB – 66.25dB, mean 46.25dB). The majority of children with conductive hearing loss had evidence of concurrent OME.”

With regards to reduced visual acuity, we based our definition on the ICD 11 criteria for 'mild & moderate vision impairment' ($>0.3\log\text{MAR} - 0.47\log\text{MAR}$ and $>0.47\log\text{MAR} - 1\log\text{MAR}$, respectively).

We acknowledge that our definition of reduced acuity ($\geq 0.3\log\text{MAR}$) was slightly more inclusive than the ICD 11 definition of 'mild vision impairment' ($>0.3\log\text{MAR}$). However, this minor discrepancy in definition led to the inclusion of only 1 child (with a visual acuity of $0.3\log\text{MAR}$).

We excluded 19 children with known ocular pathology (+/- severe visual impairment based on visual acuity); there were no additional children with severe visual impairment.

We have clarified this in the Methods section.

"Reduced acuity was defined by reduced distance acuity of the better-seeing eye $\geq 0.3\log\text{MAR}$. Acuity was assessed with glasses if worn ('habitual' state), and in the 'habitual state plus pinhole', as a proxy for full refractive correction. 19 children with known ocular pathology or severe visual impairments ($>1\log\text{MAR}$) were excluded."

We agree that it would be helpful to provide further information on the level of reduced acuity amongst the 7 children which met our criteria. We have included this information in the Results section (see below). Furthermore, we have added a sentence to explain that the reduced acuities observed are most likely to represent uncorrected refractive errors, given the technique used to assess acuity (habitual & habitual plus pinhole rather than full refractive correction) and the exclusion of children with known ocular pathology.

"Only 7 children had reduced habitual acuity in the best-seeing eye; 4 children had mild reduced visual acuity ($0.3\log\text{MAR} - 0.44\log\text{MAR}$), and 3 children had moderate reduced acuity ($0.50\log\text{MAR} - 0.7\log\text{MAR}$), with a mean acuity of $0.45\log\text{MAR}$. These reduced acuities most likely represent uncorrected refractive errors, given that acuity was not assessed with full refractive correction and the exclusion of children with ocular pathology."

Finally, we acknowledge that strabismus and amblyopia (which constitute the great majority of 'visual problems' in our study) are generally regarded as minor visual 'problems' rather than impairments, and have adjusted the terminology accordingly. However, the main aim of our study was to investigate the impact of minor, common visual and hearing problems, hence our inclusion of these conditions. We have added a paragraph to further acknowledge that they are commonly seen in the context of conditions that affect development (eg. prematurity), and thus the importance of establishing whether any educational disadvantages are attributable to these conditions per se.

"Amblyopia and strabismus are two of the most common visual difficulties encountered in childhood(10), which often occur in the context of conditions which may in themselves affect development and educational achievement such as prematurity and low birthweight(11, 12) thus rigorous adjustment for confounding factors is required to establish the functional impact of these visual difficulties in of themselves."

Comment: It would also be important to reference the reason for the thresholds chosen for attainment at KS2 and KS4 and include the normal distribution for this to compare with the study population , particularly given the differences in population from other studies as outlined in the text.

Response: We have amended the Methods section to explain that the chosen thresholds are the national benchmark measures of academic achievement.

"The national expected standard is achievement of National Curriculum level 4 or above, and we therefore used achievement of \geq level 4 at KS2 in English, Maths and Science as our three educational outcomes in primary school; performance in each subject was analysed separately."

“Achievement of 5 or more GCSEs (including Maths and English) at grades A* - C is the national benchmark measure of achievement, and we used this as our single outcome at secondary school.”

We agree that it is helpful to provide further information on how outcomes in our study compared to the national average, and have amended our Results section to include this.

“The proportion of children in this study achieving level 4 or above at KS2 exceeded the national average; 92.5% and 97.7% of children achieved level 4 or above in KS2 English and Maths tests respectively, compared to 75% and 73% of children nationally(31).”

“Overall, almost three-quarters (72.1%) of the study sample achieved 5 or more GCSEs (including English and Maths) at A*-C grade, which is significantly higher than the national average of 47.6% children achieving this standard the same year(32).”

Comment: Whilst the author has included in figure1 the information demonstrating numbers included / excluded from original cohort it would be helpful for clarity to extend this to demonstrate the numbers found to fulfil the visual/hearing criteria

Response: We have amended Figure 1 to include this data

Comment: There is a wealth of statistical data included particularly within the supplementary material which perhaps could be reduced, concentrating on the key information on the target group rather than justification of the use of regression models to increase numbers which personally I find unhelpful.

Response: We have attempted to keep the statistical data included as succinct and relevant as possible. We have attempted to achieve the correct balance based on the differing perspectives of reviewers on the level of statistical detail which should be included.

Comment: The analysis should concentrate on the actual group studied rather than using imputation which, given the unknown variables and timescales involved I would suggest may not be entirely reliable.

Response: We chose to use multiple imputation due to substantial missing data, in order to minimise potential attrition bias. This technique is well-established and enabled us to define the effect estimates more precisely (demonstrated by smaller confidence intervals for the imputed dataset analysis).

Reviewer 3:

Comment: Missing data - although it is in the supplementary material, the main paper does not appear to mention the response rate for the clinics at 7 years (or indeed for those at 8, and 9 years) where the data reported here was collected. This would be helpful to have in the main paper.

Response: We have added the attendance rates for the clinics at 7 and 8 years.

“All participating children in ALSPAC were invited to attend a research clinic at 7 years of age. 59.3% (8,299 children) attended during the period September 1998 - September 2000, of whom 98.9% (8,205 children) were eligible for inclusion (see Figure 1).”

Comment: However, it would be helpful if the authors could explain why data such as ethnicity which presumably could be reasonably easily collected at each visit was missing even when the family attended the 7 year clinic.

Response: Ethnicity data was only collected during pregnancy and at birth. The ALSPAC data dictionary (referenced in the Methods) provides a full explanation of how & when all variables were collected, with some indication of why response rates differ for different variables.

Comment: The paper does not mention specific measures of language and communication skills as far as I can see, and this is not referred to again. I wonder whether the measurement of such possible mediating factors could be mentioned in the discussion of future research directions? Might there be other possible causes such as teacher expectations (which the educational literature suggests can affect outcomes) or absences from school, which might lead to these findings?

Response: We agree that language and reading skills could be of great importance in mediating the association we have shown, but were unable to look at specific measures of these due to such high rates of missing data (explained in Methods).

We have amended the Clinical and Research Implications to highlight this as a future direction of research

“Future research involving larger numbers of participants are required to replicate these findings and elucidate further the factors mediating this association. We recommend that future investigators focus next on the role of reading and language skills as potential mediators which could explain these findings.”

Comment: If you need to shorten the paper, I wonder how much the general reader will gain from the discussion of imputation of missing values, and some of this could be included in the supplementary material?

Response: We have attempted to keep the statistical data included as succinct and relevant as possible. We have attempted to achieve the correct balance based on the differing perspectives of reviewers on the level of statistical detail which should be included.

VERSION 2 – REVIEW

REVIEWER	Reviewer name: Peter Flom Institution and Country: Peter Flom Consulting, USA Competing interests: None
REVIEW RETURNED	03-Jan-2019

GENERAL COMMENTS	The authors have addressed my concerns and I now recommend publication.
---

REVIEWER	Reviewer name: Dr Catherine Tuffrey Institution and Country: Solent NHS Trust, UK Competing interests: None
REVIEW RETURNED	07-Jan-2019

GENERAL COMMENTS	Responses to reviews all seem appropriate.
--

REVIEWER	Reviewer name: Dr Katherine Spowart Institution and Country: Greater Glasgow and clyde nhs board Competing interests: none
REVIEW RETURNED	10-Jan-2019

GENERAL COMMENTS

I can see how much work has gone in to producing this paper and revising it but I regret that I do not feel that it would add to the evidence on this subject.

While the authors have made valiant efforts to make up for the small numbers and missing data by complicated statistical methods, it seems to me that the reliability of the conclusions is questionable. I would be particularly concerned about data, which at face value, suggests that strabismus or amblyopia may be a positive advantage. This is noted as unexplained now within the paper but is of concern. Now, having information on the range of hearing level and visual acuity, which includes those with impairments recognised to put children at risk of educational disadvantage together with those with milder difficulties of unknown significance, I would question the validity of the target population. Combining these groups to increase numbers I suspect may have diluted any effect. I would also question the validity of taking a one off measure of visual acuity without refraction as an indicator of a visual difficulty - this could explain some of the 'positive effect', were these children subsequently to have been refracted and corrected.

Whilst the author has helpfully clarified some of the information within the original paper; the significant missing data with the need for complex computation, and selection criteria remain a concern.